# The Ontology of the Many-Worlds Theory

**Per Arve** 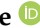

Independent Scholar, Iliongränden 21, 224 72 Lund, Sweden; per.arve@icloud.com

**Abstract:** It is shown that the wavefunction describes our observations using the postulate that relates position to the distribution $|\Psi|^2$. This finding implies that a primary ontology is unnecessary. However, what is real is not directly represented by the wavefunction but by the gauge invariants. In light of the presented ontology, Spacetime State Realism becomes not a fundamental ontology but derived.

**Keywords:** many-worlds interpretation; interpretations of quantum mechanics; wavefunction; ontology

## 1. Introduction

Schrödinger's articles in 1926 defining the wave mechanics version of quantum mechanics offered a less abstract and computationally more tractable formulation of quantum mechanics than the matrix mechanics initiated by Heisenberg. However, the wavefunction of many-particle systems depended on $N$ positions in 3-space, which defied a straightforward understanding. At the Solvay conference in 1927, Schrödinger [1] hoped that it would be possible to reformulate the theory to avoid functions of several positions in 3-space, which has not yet been found. Bohr argued in 1927 [2]:

> ... there can be no question of an immediate connection with our ordinary conceptions because the "geometrical" problem represented by the wave equation is associated with the so-called co-ordinate space, the number of dimensions which is equal to the number of degrees of freedom of the system, and, hence, in general, greater than the number of dimensions of ordinary space.

The hydrogen atom can illustrate the need for the non-relativistic quantum state to be something other than a function of a single point in space. The wavefunction for a free hydrogen atom in the ground state is a product of a center of mass function and a function of the proton and electron relative motion. Both factors are necessary to describe the physics of this system. For example, if a third particle scatters off the hydrogen atom, both factors are necessary to describe the process fully.

Bohr concluded that quantum mechanics did not constitute a description of an existing reality, but nothing more could be stated about what was going on. However, given the enormous success of quantum mechanical calculations, we should consider that the wavefunction closely mimics what is really going on. Bohmian mechanics [3–5], Everett's relative state interpretation [6–8], and wavefunction collapse theories [9] attempt to give a realistic quantum mechanical description of the physical phenomena in which the wavefunction is an integral part of the story or the whole story.

In the context of the mentioned realistic interpretations of quantum mechanics, it is still under debate what kind of entity the wavefunction is. The problem that the founders of QM faced remains. The number of real variables is $3N$ for $N$ particles, but the physical space we experience is three-dimensional. Maudlin [10,11] has been skeptical that a function of so many variables can give the full account of what is going on in 3-space. Albert [12] has taken seriously that the dimension of the domain implies that the dimension of physical space is $3N$. The wavefunction becomes a field in that space. He argues that

the Hamiltonian implies a 3-dimensional emergent structure corresponding to the 3-space we experience. This is also the view Ney has taken [13], but she differs in precisely how the three-dimensional structure is extracted. The alternative is to take the 3-space as fundamental and to accept that QM introduces a physical quantity that depends on several points in the 3-space. This has been advocated by Lewis [14], Ney [15], and Chen [16].

Section 2 discusses that the wavefunction is a function of several points in space. How such a wavefunction can describe what we experience is elaborated on in Section 3. Section 4 presents the reality the wavefunction implies. Wallace and Timpson have previously suggested what can be understood as the reality of the wavefunction, which is discussed in Section 5. Section 6 summarizes the findings.

## 2. The Configuration Space for $N$ Point Particles and the Wavefunction

$N$ points give the configuration of $N$ classical point particles located somewhere in 3-space. The space of all possible configurations, the configuration space, is the set of all possible configurations of $N$ points, which will here be denoted by C($N$). The symbol $\overline{\mathbf{x}}$ denotes an element in C($N$). Figure 1 shows an element of C(5). Even though $3N$ real numbers give a configuration, it is not correct to state that C($N$) equals $\mathbb{R}^{3N}$. C($N$) contains a structure that is missing in $\mathbb{R}^{3N}$. For example, the coordinates for all points are relative to the same coordinate system in 3-space, and there is a distance measure between points

$$r = \sqrt{(x_i - x_j)^2 + (y_i - y_j)^2 + (z_i - z_j)^2}, \tag{1}$$

where $i$ and $j$ are particle indicies. This kind of distance is available in C($N$) but not in $\mathbb{R}^{3N}$. Simply, C($N$) $\neq \mathbb{R}^{3N}$. One might abstract away those features of C($N$), that is to ignore them, and replace C($N$) by $\mathbb{R}^{3N}$, but only when those features are irrelevant. (If there were one bowl with five apples and another with five oranges, you might use the notion fruits, which is an abstract notion relative to apples and oranges, and say that five apples =(fruits) five oranges. In doing so, we ignore the difference between apples and oranges. In some circumstances, this omission would be fine; in others, less so).

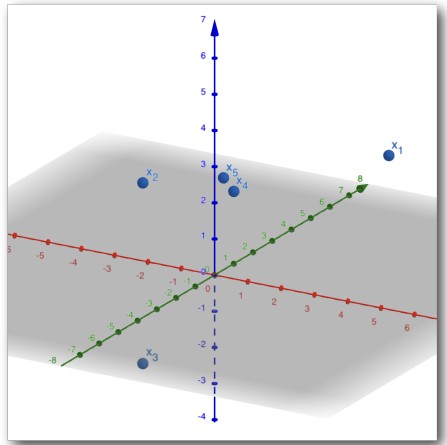

**Figure 1.** $\overline{\mathbf{x}} = \mathbf{x}_1, \mathbf{x}_2, \mathbf{x}_3, \mathbf{x}_4, \mathbf{x}_5 \in$ C(5).

When physicists perform calculations of quantum many-particle systems, the wave function, $\Psi(\overline{\mathbf{x}}, t)$, is a mapping $\Psi : $ C($N$) $\to \mathbb{C}^P$, where $P$ is the dimension of the combined spin space and $\mathbb{C}$ is the complex numbers. (As $\mathbf{E}(\mathbf{x}, t_0)$ gives the configuration of the electric field in space at time $t_0$, $\Psi(\overline{\mathbf{x}}, t_0)$ is the configuration of the quantum state on C($N$). Hence, we can view C($N$) as the space on which the quantum state is configured, its configuration space.) This definition of the wavefunction is used, together with the standard method for comparing with measurements, to achieve great success. From this fact about how quantum mechanics is actually applied, it is surprising that so many physicists and philosophers of physics take the domain of the wavefunction to be the unstructured $\mathbb{R}^{3N}$ rather than

C(*N*). For example, Maudlin [10] writes, "The wavefunction is something that evolves in a very, very, very, very high-dimensional space", and then continues that "there is no low dimensional space at all" as a description of the domain of the wavefunction.

In the quest to find the foundations of QM, the actually used theory is to be analyzed. The straightforward understanding of the world that QM describes is that 3-space is fundamental and that the wavefunction is a function of *N* points in 3-space. Such a mathematical entity has been denoted "poly-wave" or "polyadic field" by Forrest [17] and in the context of pilot wave theory "multi-field" by Hubert and Romano [18]. From QFT, we are used to *N*-point functions (correlations) $\langle |\phi_1(x_1)\phi_2(x_2), \ldots, \phi_N(x_N)| \rangle$, where $\phi_i(x_i)$ is some local operator at some position and time $x_i = (\mathbf{x}_i, t_i)$. From such *N*-point functions, we can arrive at the non-relativistic quantum mechanics from QFT. This relation shows that we should take the 3-space as fundamental. However, the degrees of freedom a physical system possesses are a fundamental property of that system, so the space of the 3*N* degrees of freedom is fundamental in a universe of *N* particles. Henceforth, functions of *N* points in space will generally be called *N*-point functions.

Albert and Ney assume that the 3*N* dimensional space is fundamental. They use arguments about the dynamics or invariances of the systems to argue that 3-space emerges. Interactions depend on the Expression (1). Its symmetries are the symmetries of the 3D space that emerges, and *r* is the distance measure that makes it into a Euclidean space. The arguments also imply that the wavefunction variables are divided into triples that correspond to a point in this space. One might argue that the implied 3D space is a nomic structure as a fundamental feature of the interactions implies its existence. The fact that this 3D space is present for any value of *N* implies that 3D space is more fundamental than 3*N* space. That Albert and Ney posited the 3N space to be fundamental does not exclude that their investigations will lead to the 3D space also being fundamental and even more so than the 3N space. We end up here in that the wavefunction variables correspond to *N* points in a 3D Euclidian space, which we have to identify with the ordinary 3-space we observe.

To conclude, the wavefunction domain has the properties of C(*N*): either we posit it or we discover it (The notion of emergence is misleading here. If we start from one space being fundamental and then find the existence of another fundamental space, we should not think of the second space as emergent. We simply discovered something we were unaware of).

## 3. The Wavefunction Description of What Is Going on in 3-Space

We have established that the wavefunction domain is a set of geometrical figures in 3-space, *N* points in 3-space, to a complex linear space. This structure proves that the wavefunction is a structure in 3-space [19]. It remains to be shown that it can explain what we observe.

Everett's vision that the unitarily evolving universal wavefunction can describe every aspect of the physical world needs a statement about the physical significance of the value of Ψ. In particular, the wavefunction has to contain information about where everything is located. As the wavefunction is a distributed object, the position of a particle (or particles) can only be given by a distribution. As the theory does not contain any point-like entities, this distribution is not a probability distribution of the point the particle(s) location. We have to abolish the traditional view of the location of particles as being points. Positions in Everett's Quantum Mechanics (EQM) necessarily have to be a distributed quantity. Arve [20] formulated in a postulate (EQM1) what that distribution should be. The position $\bar{\mathbf{x}}$ and spin value *a* of the particles is given by the distribution

$$\rho(\bar{\mathbf{x}}, a) = |\Psi_a(\bar{\mathbf{x}})|^2, \tag{2}$$

which is called the presence distribution. It has also been called "measure of existence" [21] and "partial instantiation" [13], which seems to possess a meaning similar to presence. Greaves [22] has argued that $\rho$ is the "caring measure," which fails to give a general

understanding of the quantity, as it is only relevant for agents making decisions. Position, existence, and instance are notions that we are not used to having a gradual character in the sense suggested, but Everett's vision implies that we accept that at least one of these notions to be a distributed quantity given by $\rho(\bar{\mathbf{x}}, a)$. In [20], the quantity

$$P = \sum_a \int_{V_1} \cdots \int_{V_N} \rho(\bar{\mathbf{x}}, a) d^{3N}x \qquad (3)$$

is called the presence in $V_1 \times \ldots \times V_N$. The article proved that an observer should have the same expected relative frequencies as if the Born was applicable because we should expect to find ourselves in a situation associated with high presence (In turn, this implies that a rational agent should make decisions as if the Born rule is true).

Consider a scattering described by two initial wave packets that collide. One of them, the target, initially has zero group velocity, while the projectile has a finite and known group velocity. After the collision, the combined system is entangled. Assume that an array of detectors is set up at a macroscopic distance away from the collision region at positions covering the angles where the projectile or target will have an appreciable presence. Due to the agreement with the Born rule, we know that EQM describes the frequencies with which detectors measure the projectile and target system at different angles, including the correlations between projectile and target. We have a description of how the combined wavefunction of the target and the projectile evolve in 3-space in agreement with observations. This description contains correlations between the projectile and the target. The description cannot be separated into one description of where the projectile is located and another description of where the target is located. We can calculate what is called the marginals in the context of probabilities to get the distribution of one of the systems, corresponding to measuring only one of the particles. In contrast, the entire presence distribution is necessary to get the correct theory for coincidence experiments.

For an atom, a molecule, or any other bound system free from external forces, the wavefunction is a superposition of states of the type (A subsystem of the world should strictly be given by density matrix. The columns of the matrix are then such superpositions)

$$\psi_{CM}^{(i)}(\mathbf{x}_{CM}) \psi_i(\bar{\mathbf{x}}_i). \qquad (4)$$

The center of mass wavefunction $\psi_{CM}^{(i)}(\mathbf{x}_{CM})$ can be of any shape that we can consider for a free non-relativistic point particle, and its absolute squared gives the position distribution of the center of mass. The intrinsic state $\psi_i(\bar{\mathbf{x}}_i)$ absolute squared gives the position distribution of the parts relative to the center of mass. From the intrinsic states, we can get the excitation spectra and all matrix elements related to the coupling to an external probe. In atoms with several electrons, there are correlations corresponding to entanglement between the electrons.

The fact that the domain of the wavefunction is C($N$) implies that it describes something in 3-space. From the considerations above, the wavefunction clearly describes what is going on in a scattering event and all the structures of atoms and molecules that we can have precise knowledge about according to QM. That the wavefunction is a function of not just one but several points in 3-space made Bohr claim that it does not describe what is going on. Our experience from QM calculations of various physical systems combined with the postulate EQM1 proves the ability of the wavefunction viewed as a $N$-point function to describe what is going on in 3-space.

So far, we have assumed the existence of macroscopic objects like detectors, tables, and other objects. These are nothing but large generalized molecular structures where atoms have relatively well-defined relative positions. That macroscopic objects have well-defined positions relative to each other is guaranteed by decoherence, which is present under normal circumstances.

Decoherence is also vital in splitting a world into several branches in an experiment where the wavefunction of the measured system contains many values of the measured

quantity. In this quantum mechanical description of the measurement process and in many other situations where entanglement is a prominent feature, it is vital to view the wavefunction as a function of several points in 3-space.

## 4. Proposed Ontology

As argued in [20], the ontology ought to be gauge invariant as the gauge choice has no physical consequences. This result only depends on that the Hamiltonian is a kinetic energy term and a potential energy, which is a function of positions and spin.

A gauge change amounts to adding to the vector potential of particle type $l$ the field $\Delta \mathbf{A}_l(\mathbf{x})$. Here, all particle types have their gauge fields, also neutral particles. For charged particles the gauge field includes the value of the charge. The product of the charge $q_l$ and magnetic field $\mathbf{B}$ are given by $q_l \mathbf{B} = \nabla \wedge \mathbf{A}_l$. For neutral particles $\nabla \wedge \mathbf{A}_l = 0$. The gauge change is culs free, $\nabla \wedge \Delta \mathbf{A}_l = 0$ and changes the wavefunction,

$$\Psi(\overline{\mathbf{x}}) \rightarrow \exp\left( \frac{-i}{\hbar} \sum_{k=1}^{N} \int^{x_k} \Delta \mathbf{A}_{l_k}(\mathbf{x}'_k) dx'_k \right) \Psi(\overline{\mathbf{x}}). \tag{5}$$

The gauge-independent quantities

$$\rho(\overline{\mathbf{x}}) = \sum_a |\Psi_a(\overline{\mathbf{x}})|^2, \quad \mathbf{j}_k(\overline{\mathbf{x}}) = \frac{1}{m} \mathrm{Re} \sum_a \left[ \Psi_a^*(\overline{\mathbf{x}}) \left( \frac{\hbar}{i} \nabla_k + \mathbf{A}_{l_k}(\mathbf{x}_k) \right) \Psi_a(\overline{\mathbf{x}}) \right], \, k = 1, \ldots, N, \tag{6}$$

and the total spin state Hilbert space ray, $S(\overline{\mathbf{x}})$, is the ontology related to the wavefunction. Given the vector fields, a global phase choice, and $\mathbf{A}_l(\mathbf{x})$, the wavefunction can be derived from these gauge invariant quantities. Note that only if the N point functions $\rho$, $\overline{\mathbf{j}}$, fulfill a certain condition [23] can they correspond to a wavefunction. But when they are derived from a wavefunction, $\rho$, $\overline{\mathbf{j}}$, $S$, together with the set of gauge fields $\{\mathbf{A}_l\}$ they give the wavefunction uniquely except for the global phase choice. Vaidman has denoted the quantity $\rho(\overline{\mathbf{x}})$ by "measure of existence", which suits its ontological character. The spin quantity $S(\overline{\mathbf{x}})$ is gauge invariant and, as far as is known, does not contain any superfluous degrees of freedom.

That the interactions are local favors strongly that the Schrödinger equation is written in the spatial basis. Thus the quantum state is naturally represented by the wavefunction $\Psi(\overline{\mathbf{x}})$ due to the locality of the interactions. That feature also implies the gauge invariance and that the ontology contains the quantities given here. What would be the ontology if the interactions were non-local is something we need not be concerned with because we have no understanding of what such a world would be like, nor do we have any good reason to study such a world.

It is often stated that the fundamental understanding of the quantum state is that it is a vector in Hilbert space. Without further specifications, a statement to this end, the quantum state is empty of physical significance, as the abstract Hilbert space is a purely abstract mathematical entity like the natural numbers. A specific number, e.g., 5, says nothing about the physical world without a context that tells what the number stands for. However, there are concrete Hilbert spaces, specifically $L_2[C(N) \rightarrow \mathbb{C}^P]$. This Hilbert space is the correct concrete one to which the wavefunction belongs. For a Hilbert space representation to describe some features of our world, there has to be one basis directly related to the fundamental description, and for any other basis state, we have to express it in terms of this fundamental basis. Rewriting any state or relation on a non-fundamental basis is nothing but a mathematical transform. As such, it can be beneficial for various considerations. When it comes to quantum mechanics, a warning is prudent. Any such transform is gauge-dependent. If the gauge is changed, the form of the transformed expressions should change to preserve its physical meaning.

## 5. Spacetime State Realism

Wallace and Timpson [24] have offered an alternative to wavefunction realism which they call Spacetime State Realism (SSR). Their proposal is put forward as an alternative to wavefunction realism, which they have criticized.

They focus on giving the quantum state an understanding in terms of subsystems localized in 3-space. The density matrix gives the quantum state in a spatial region $\Delta$, which is obtained by using a Hilbert space basis divided into states with support inside and support outside $\Delta$. This construction leads to a varying particle number inside the region. In the case of QFT, they let (the action of) local operators restricted to the region define the state in the region.

The authors give a minimal argument about how they thought this could be done. Inside $\Delta$, the wavefunction was considered a superposition of the products of single-particle states, with support only inside the region $\Delta$. For the single-particle wavefunctions to have support inside a region in space, they have to be functions of a position in 3-space, which implies that the universal wavefunction is a function of many points in 3-space. However, the authors never comment on how the wavefunction is related to 3-space. For SSR to give the features that the authors advocated to be its advantage over $\mathbb{R}^{3N}$ wavefunction reality, the wavefunction has to be an entity in 3-space. However, from Section 3, it is clear that wavefunction is an entity in 3-space when it is taken to be a function of positions in 3-space. The present analysis has closed a gap in the argumentation for SSR.

Wallace and Timpson argued that the ontology ought not to be one big system with no subsystem decomposition because we would have only a single property bearer which "would lack sufficient articulation to give the physical meaning of what was presented". This assertion is not warranted. Taking the universal wavefunction, or rather the position distribution $\rho(\bar{\mathbf{x}})$, the current $\bar{\mathbf{j}}(\bar{\mathbf{x}})$, and the spin state $S(\bar{\mathbf{x}})$ as the fundamental ontology, there would be derived local ontological features in terms of the density matrices that SSR is based on.

Wallace and Timpson recognize that the main drawback of SSR is that it separates into local regions, though the wavefunction is non-separable. This feature is a grave problem that disqualifies a set of local density matrices from being the fundamental ontology as it cannot represent all physical features. SSR leaves out the entanglements of entities in different regions of their space division. A fundamental ontology must be able to represent all physical relations and effects.

Wallace and Timpson described a version of SSR for QFT, which focuses on the algebra of local operators. As the algebra is only local, the algebraic relations between operators at widely different positions and their expectation values were not included. Thus, the entanglements of entities at different spatial regions are omitted here as well. This version is equally unfit to constitute the fundamental ontology as the non-relativistic case. Additionally, Swanson [25] has pointed out technical difficulties in the approach to QFT SSR.

One of the points of criticism against the $\mathbb{R}^{3N}$ wavefunction realism was that relativistic QFT gives a very different picture in which particles are emergent and not fundamental. This criticism implies that any version of wavefunction realism where the particle number $N$ is fundamental is mistaken about what is fundamental. However, it is a legitimate investigation to find out what is real within a theory like non-relativistic QM that describes so much of the world around us. Relativistic QFT can be seen to be more fundamental, but it is hardly the ultimate theory of the physical world. It is indeed vital to investigate the theories we have. The principle that the ontology should be given by the gauge invariant entities, as advocated in Section 4, will probably also produce a good understanding of the QFT ontology.

Maudlin has criticized SSR with that the density matrix will contain information from the many worlds created since the Big Bang, which is present in the region. Maudlin argues that the density matrix will essentially be a continuous distribution containing no discernible information. In particular, this is an argument against the possibility of dividing

the density matrix into a sum of quasi-classical worlds, which Wallace and Timpson claimed to be possible.

The most severe criticism against SSR is that it never explains how any aspect of the physical reality is connected to the amplitudes that enter the construction of the density matrices. For example, this could be achieved by statements similar to EQM1, but Wallace and Timpson failed to see its necessity. It can be added that the same criticism applies to the argumentation for Everett's ideas found in Wallace's book [8]. No interpretation of the wavefunction amplitude is given so that the patterns in the amplitude can be interpreted in terms of physical objects.

*Albert's Narration Paradox*

David Albert [26] has found a consistency problem for the non-relativistic QM description of the following scenarios. The discussion of Albert, called a narratability problem, will follow the presentation in [24]. Two spin-1/2 particles at a distance from each other and the spins form together a spin singlet. In the first scenario, nothing happens. In the second scenario, both spins are flipped simultaneously such that $|\uparrow\rangle \to |\downarrow\rangle$ and $|\downarrow\rangle \to -|\uparrow\rangle$. Then the spin singlet state is unchanged afterward,

$$|\uparrow\rangle|\downarrow\rangle - |\downarrow\rangle|\uparrow\rangle \to -|\downarrow\rangle|\uparrow\rangle + |\uparrow\rangle|\downarrow\rangle. \tag{7}$$

In both scenarios, the spin state of the combined system is always in a spin singlet. However, from the point of a moving frame, the changes of the two spins will not be simultaneous. The state between the two changes might then become

$$|\downarrow\rangle|\downarrow\rangle + |\uparrow\rangle|\uparrow\rangle. \tag{8}$$

In the original frame of reference, there was no period in which the state was in a spin triplet which we have in the moving frame. Wallace and Timpson state that the sequence of states in the moving frame, $\Psi'(t)$, is not a mere redescription of the state sequence in the original frame $\Psi(t)$. They further conclude that the sequence of states demonstrates that $\Psi(t)$ cannot be regarded as fully describing the properties of the system.

There are a couple of problems with the description and the conclusions. That systems might seem qualitatively different in frames moving with respect to each other is well-known to seem paradoxical, but we have to accept the consistency of the theory. For example, in one frame, a train might, for a moment, be entirely inside a tunnel, while in another frame, it is never the case. The difference between the frames in the train "paradox" can easily be resolved. In both frames, consider the events that the back of the train enters the tunnel and the front of the train coming out of the tunnel. For the effects of the triplet state to become apparent, consider simultaneous measurements in the moving frame of the two spins. The result of such measurements in the basic direction will demonstrate that the spins have equal direction. In the non-moving frame, these measurements will happen at different times. One will be before the spin flips and the other one afterward. No surprise that the spins will be measured to have the same direction as is the case for the triplet state. There is no more of a problem or a paradox here than in the case of the train and the tunnel. A shortcoming of the spin scenarios is that it takes some time for the spins to change direction. There also needs to be an apparatus to flip the spins, which should be included in the quantum description. In Albert's version, two additional particles are involved in flipping the spins. This more complicated situation requires a lengthy discussion which we will not embark on here.

The conclusions that Wallace and Timpson made are not warranted. The descriptions that the $\Psi(t)$ or $\Psi'(t)$ give are in as much agreement as is necessary and allowed by the theory of relativity.

## 6. Summary

The wavefunction is a function of $N$ points in 3-space; the domain is C($N$). This domain implies that the wavefunction describes things happening in 3-space. By examining a couple of example systems, it was shown that the wavefunction could describe our observations. For that end, the postulate EQM1 is necessary. Only gauge invariant quantities can be ontic. The sufficient and minimal ontic components are the presence $\rho(\bar{\mathbf{x}})$, the current $\bar{\mathbf{j}}(\bar{\mathbf{x}})$, and the total spin state $S(\bar{\mathbf{x}})$. Bohr's pessimism about the possibility that the wavefunction describes "our ordinary conceptions" has been proven unwarranted. The success of describing our observations of physical systems and experiments with only the wavefunction gauge invariants demonstrates that a primitive ontology is not necessary.

The previously proposed SSR is problematic. Its authors' arguments against wavefunction realism were directed against the version in which the wavefunction domain was taken to be $\mathbb{R}^{3N}$, for which the ordinary 3-space is not clearly present. However, SSR is founded on the view that the wavefunction domain is C($N$), but Wallace and Timpson never discussed the possibility that it defines what is real. Instead, they defined the density matrix for the subsystem being a region in space to be what is real. Then the information about entanglement with the world outside the region is lost, which renders the ontology incomplete. The most devastating problem of SSR is that the wavefunction is not given any physical significance, rendering the density matrix meaningless.

**Funding:** This research received no external funding.

**Institutional Review Board Statement:** Not applicable.

**Informed Consent Statement:** Not applicable.

**Data Availability Statement:** No new data were created or analyzed in this study. Data sharing is not applicable to this article.

**Acknowledgments:** Discussions with Chris Stoica and David Wallace are acknowledged.

**Conflicts of Interest:** The author declares no conflict of interest.

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
