# Peer review of "The Ontology of the Many-Worlds Theory"

_quantumrep, doi:10.3390/quantum5010015_

Round 1
Reviewer 1 Report
The manuscript by Per Arve concerns the role and physical status of the wave function in the many-world interpretations.
The author compare several positions due to Albert and others where the wave function is physically either defined in the 3N-dimensional configuration space or alternatively in the 3D space. The view taken by the author is indeed that the fundamental space is our 3D space where the wave function acting as a multi-filed Psi(x1,x2,…xN,t) defines a kind of correlation between various points: ”The wave function is a set of geometrical objects in 3D, N points in 3-space, to a complex linear space”. The paper is well written and clear and the thesis is well defended.
Two aspects could be improved:
-First there is the issue of QFT where the configuration space dimension is infinite (uncountable?). It is not clear in the present version if the same approach as used with non relativistic quantum mechanics could be applied or extended in QFT without trouble.
- It is not very clear if there is really any difference between an approach where the wave function is living in a 3 or 3N spaces. In the end the quantum predictions are the same. To say that this is a different ontology is for me just a figure of rhetoric, ie., lietelling that you prefer to use a Lagrangian more than an Hamiltonian. It seems to me that to make a difference between two mathematically equivalent approaches one must include external elements. In other words one must beyond a pure formalism where there is always a one to one transformation between the two pictures in the 3 or 3N spaces. Perhaps the author could comment a bit on that issue.
Considering these remarks I anyway recommend the paper for publications in quantum reports.
Ps: It seems that reference 10 of Maudlin’s work is not complete: the ref of the book is missing.
Author Response
Thanks for noticing the incompleteness in ref 10. Will be fixed in a revised version to be submitted.
Also for QFT it is that case guage choices don't affect anything of relevance. For the case of QED it is pretty straightforward to identify the ontology, though to present that result goes well beyond the intentions with the present article. However, the case of non-abelian gauge theories aren't yet studied, why a discussion of QFT:s will have to wait for the completion of that investigation.
To my knowledge, all calculations of physical quantities that ever have been done involving many-particle wavefunctions have taken the wavefunction to depend on N points in 3-space. That is, the domain has be taken to be what I denote C(N). I am not aware of any description of how to calculate any physical quantities by taking the domain to be R3N, nor can I think that it could be possible. In any calculation we take the variables to be triples consisting of an x-coordinate in 3-space, an y-coordinate in 3-space, and an z-coordinate. If we do that, we have assumed that the domain is C(N), possibly without realizing it. Taking the domain to be R3N, then there is no link between the wavefunction variables and 3-space. The set R3N contain no structure other than that every element consists of 3N real numbers. If you add some more structure to that space, it is something else than R3N. Note that all physics textbooks take the domain to be C(N), though most never explicitly mention what the domain is. However, in "Quantum Theory of Many-Particle Systems" by Fetter and Walecka says "The quantity xk denotes the kth particle, including the spatial coordinate xk and any discrete variables as the z-component of the spin ...".
Reviewer 2 Report
The paper presents a relevant proposal for an ontology of the Many-Worlds interpretation, addressing one of the main (philosophical) issues of the MWI.
While I like that the paper is concise -- and understand that more details are provided in Arve (2020) -- I find the present discussion, at times, too quick. Several aspects of the proposed ontology remain somewhat unclear to me.
Is the proposal, in the end, that \rho and j should be interpreted as multi-fields on three-dimensional physical space? (On multi-fields, see also Hubert and Romano, 2018, "The wave function as a multi-field.")
What is the meaning and status of "the total spin state Hilbert space ray" that appears like an additional ontological commitment (p. 5)?
In any case, the claim that "given the vector fields, the wavefunction can be derived from these gauge invariant quantities" seems wrong. (In particular, it's not possible to recover the Schrödinger equation for \Psi from the Madelung equations for \rho and j; see Wallstrom, Phys. Rev. A 49, 1994).
How to understand the claim that "we should expect to find ourselves in a situation associated with high presence"? Will we not necessarily split into a great (or maybe indefinite) number of descendants that all co-exist with various degrees of presence (whatever that means)?
The paper also announces to show that the Spacetime State Realism proposed by Wallace and Timpson is problematic, but most of Section 5 seems to respond to their criticism against wave function realism. The points explicitly raised against SSR itself strike me as a bit unfair, although I'm far from an advocate of this view.
"... there is a related point of inconsistency in SSR. It is only possible to consider the density matrix of a limited region of space if the quantum state already has a clear relation to 3-space, which implies that the wavefunction domain is C(N)."
In the relativistic context, SSR does not try to assign states to bounded regions of 3-space but of 4-dimensional spacetime.
"The most severe criticism against SSR is that they never give how any aspect of the physical reality is connected to the density matrices."
Wallace and Timpson say a little bit about it, and Wallace says a lot more in his "Emergent Multiverse" (2012). I agree that the account remains vague, but it doesn't seem fair to suggest that there's none.
Author Response
Interpreting quantum mechanics is a mutual task for physicists and philosophers. As a physicist, I am curious about what in the wavefunction are the fundamental physical entities, which corresponds to asking what the ontology is.
The quantities \rho and j are functions of N-points in space (N-point function), which follows from that the wavefunction is that. The reviewer has correctly directed my attention to the fact that Hubert and Romano are the originators of the term ”multi-field”. However, I have now decided not to use the term ”multi-field” but for a brief mention of Hubert and Romano's work. Their work is very focused on pilot wave theory. An alternative that they rejected was the polyadic wave, which Forrest introduced. Rather than switching to ”polyadic wave”, I have decided to use the term ”N-point function” in a revised manuscript. This term has been established in QFT since much before the other two. There is no reason to estrange physicists by inventing a new term when an established term is available.
As stated in the manuscript, the total spin-state Hilbert space ray is a part of the ontology. In Everett’s quantum mechanics, it is evident that the spins are as real as any spatial quantities. Perhaps someone coming from pilot wave theory would be surprised, where spins are not included in the primitive ontology but only appears in the ”secondary” ontology, the wavefunction. Here, there is no primitive ontology.
My claim that the ontology and the gauge field give the wavefunction is correct. Wallstrom’s work, which I am well aware of, does not conflict with my statement. He gave the condition for when \rho and j can correspond to a wavefunction, but here we know that they do, so the condition is satisfied. I have added a note in the revised manuscript addressing this question.
The reviewer asks what I mean by "we should expect to find ourselves in a situation associated with high presence.” To give a short answer here, if we face a future with a set of branches, we will become distributed with varying presence in the different branches. We should expect to see things as they are in the individual branches depending on the presence in that branch. However, I expect this answer will provoke further questions. The reader should consult my article on these matters Found Phys 50, 665–692 (2020).
Inspired by the reviewer’s criticism, I rewrote the revised manuscript's section on SSR. It is now clearly focused on the relationship between SSR and the ontology advocated in my article.
However, the criticism against SSR that it gives no connection with physics stands. Mathematical equations need a statement that relates quantities to observed entities. The authors gave no physical meaning to the amplitudes in the density matrix. I have not found any statement in Wallace's book that fills this gap. In discussions, Wallace has shown no willingness to commit to any general statement that links the amplitude to something in our observable reality. Many of the critics of EQM/MWI, e.g. Bell, Maudlin, Kent, and Fuchs, complain that the theory is empty of content
Reviewer 3 Report
The refereed paper is called "The ontology of the many-worlds theory", but I did not find there any connections with the many-worlds interpretation of the quantum mechanics. The author thinks that the main problem of the interpretation of quantum mechanics is connected with the fact the wave function of the N-particle system is defined in 3N-dimensional coordinate space and not in the 3-dimensional space. Then the author consider the dynamics of the system of N particles and tells that it should be described in terms of gauge-invariant quantities, while what is considered here is not a gauge field theory but a simple non-relativistic dynamics (it is not clear classical or quantum) of the N particles. I did not find in the paper any new results and the presentation of the ideas here is rather confusing. I do not recommend this paper for the publication.
Author Response
The reviewer seems not familiar enough with EQM/MWI to know that it is a theory where the state is given by nothing but the (universal) wavefunction which is branching at certain events. The reviewer is also unfamiliar with the guage degree of freedom that exists in non-relativistic quantum mechanics, often presented in the first or second technical course on the subject.
It is not meaningful to further consider this review, considering the reviewer's lack of understanding of the subject.